# Therapeutic Effects of Mutian^®^ Xraphconn on 141 Client-Owned Cats with Feline Infectious Peritonitis Predicted by Total Bilirubin Levels

**DOI:** 10.3390/vetsci8120328

**Published:** 2021-12-13

**Authors:** Masato Katayama, Yukina Uemura

**Affiliations:** Bloom Animal Hospital, Kajiyama 1-10-32, Tsurumi, Yokohama City 230-0072, Kanagawa, Japan; marble1993.22@gmail.com

**Keywords:** FIP, cats, nucleoside analogue, therapy, total bilirubin, prediction

## Abstract

Feline infectious peritonitis (FIP) is a fatal disease caused by feline coronavirus or its variant, referred to as the FIP virus. Recently, favorable treatment outcomes of the anti-viral drug Mutian^®^ Xraphconn (Mutian X) were noted in cats with FIP. Thus, the therapeutic efficacy of Mutian X in cats with FIP must be explored, although the predictors of therapeutic success remain unknown. In the present study, we administered Mutian X to 141 pet cats with effusive FIP following initial veterinarian examinations. Of these, 116 cats survived but the remaining 25 died during treatment. Pre-treatment signalment, viral gene expression, and representative laboratory parameters for routine FIP diagnosis (i.e., hematocrit, albumin-to-globulin ratio, total bilirubin, serum amyloid-A, and α1-acid glycoprotein) were statistically compared between the survivor and non-survivor groups. The majority of these parameters, including hematocrit, albumin-to-globulin ratio, serum amyloid-A, α1-acid glycoprotein, and viral gene expression, were comparable between the two groups. Interestingly, however, total bilirubin levels in the survivor group were significantly lower than those in the non-survivor group (*p* < 0.0001). Furthermore, in almost all surviving cats with effusive FIP (96.6%, 28/29), the pre-treatment total bilirubin levels were below 0.5 mg/dL; however, the survival rate decreased drastically (14.3%, 1/7) when the pre-treatment total bilirubin levels exceeded 4.0 mg/dL. Thus, circulating total bilirubin levels may act as a prognostic risk factor for severe FIP and may serve as the predictor of the therapeutic efficacy of Mutian X against this fatal disease.

## 1. Introduction

Feline infectious peritonitis (FIP) is a fatal disease caused by a variant of feline coronavirus, referred to as the FIP virus (FIPV) [1,2,3]. At the early stages of the disease, FIP-infected cats present with non-specific symptoms, including recurrent fever, vomiting, diarrhea, and progressively worsening weight loss, and the characteristic signs of overt FIP manifest as the disease progresses [1,3]. FIP is classified into effusive, non-effusive, or a combination of both types [3,4]. The effusive type of the disease is characterized by the accumulation of ascites or pleural fluid, while the non-effusive type is characterized by the granulomatous involvement of several organs, central nervous system symptoms, or ocular manifestations; the mixed type is characterized by all these [1,4]. Currently, the antemortem diagnosis of FIP is difficult, and no non-invasive confirmatory test is available to diagnose any type of FIP, effusive or not; nonetheless, diagnosis can be made with reasonable certainty based on the results of physical examination, clinicopathological tests, antibody titers for feline coronavirus, and viral gene expression analysis using polymerase chain reaction (PCR) of biological specimens, including blood, ascites, or pleural effusions [4,5].

Currently, there is no widely available and legal effective therapy for FIP, and only symptomatic treatments can be administered, including the drainage of ascites or pleural effusions, administration of steroids or antibiotics, and supplemental nutrition, all aimed at prolonging survival [5,6]. Although several studies have reported the use of interferon-ω, xanthine derivatives, and polyprenyl immune stimulants as anti-viral agents, their clinical advantages have never been verified [7,8,9]. In addition, itraconazole, chloroquine, and cyclosporin A have been tested as therapeutic agents, although their side effects and poor outcomes are unignorable [10,11,12,13]. Consequently, given the lack of investigations on prophylactic agents, avoiding group hoarding or reducing stress of the affected cats remains the residual measure to relieve symptoms [1,4,5,6].

In 2018, the suppressive effects of the nucleoside analog GS-441524 on FIPV were demonstrated [14]. The phosphoramidate prodrug of GS-441524, designated as GS-5734 (Remdesivir; Gilead Sciences), could inhibit the replication of several taxonomically diverse RNA viruses, such as Ebola virus [15]. The exceptional efficacy of GS-441524 against naturally occurring, non-neurological FIP was proven based on over 80% recovery from disease onset, although the mortality rate of FIP remained 100% despite previous therapies [16]. Moreover, GS-441524 administration at higher doses was effective in cats with ocular and neurological FIP [17]. Meanwhile, field testing of GC376—a novel 3C-like protease inhibitor with potential therapeutic effects against FIP—demonstrated that this drug may require higher doses and longer duration for FIP treatment, and the cure rate was expected to be lower [18]. Furthermore, given the potential medicinal effects of adenosine nucleoside analogs, such as GS-441524, on FIP, their orally administrable products for veterinary use were developed by a Chinese company [19] and are now available in Asian countries, including Japan.

To this end, in the present study, we describe our experience of drastic symptomatic improvement and apparent life prolongation following the administration of a novel adenosine nucleoside analog to cats with effusive FIP. We also propose possible prognostic indicators to address the decision to treat the affected cats.

## 2. Materials and Methods

### 2.1. Therapeutic Agent and Administration

As a therapeutic agent for FIP, we used Mutian^®^ Xraphconn (Mutian X), which was developed by Mutian Life Sciences Co., Ltd (Nantong, China) in 2019. Currently, two types of drug products, capsulized for oral administration and filled in vials for subcutaneous injection, are supplied by the manufacturer [19]. The anti-FIPV mechanism of the adenosine nucleoside analog, the key active ingredient of Mutian X, remains unknown. Mild digestive organ abnormality (specifically diarrhea) and liver dysfunction were observed following drug administration, although there were no severe adverse reactions [19]. We administered Mutian X through the oral or subcutaneous route to 141 cats diagnosed with effusive FIP at the Bloom Animal Hospital (Tsurumi, Yokohama, Japan) between June 2019 and December 2020. The drug was administered following the manufacture’s recommendations. Briefly, immediately after the first diagnosis of FIP, Mutian X was administered orally or injected subcutaneously at 100 mg/kg to all cats (100 mg formulation includes approximately 5 mg of the active ingredient of adenosine nucleoside analog, designated MT0901), and the course was continued between days 0 and 84 [19]. The capsulized form is administered orally. Alternatively, the subcutaneous form can be injected at the same dosage, if oral administration is difficult because of gastrointestinal dysfunction and inability to absorb nutrients due to the onset of FIP. Both drug administrations must be performed on a daily basis and on an empty stomach [19]. At our hospital, each cat was followed up for 3 months post the standard drug administration course of 84 days, and the stable animals without apparent quality of life impairment were judged to be in remission.

### 2.2. Animals and Diagnosis

A total of 141 cats were diagnosed with effusive FIP upon initial veterinary examinations at our hospital between June 2019 and December 2020. Only cats with effusive FIP were included in the present study since the required number of animals to form a satisfactory cohort for statistics could be reached. Unfortunately, however, the number of cats with non-effusive or mixed FIP was not sufficient for statistical analyses. An additional 28 cats were transferred from other institutions due to the suspicion of FIP but were confirmed to not present with the disease at our hospital within the same period (cats without FIP). Finally, 169 animals were included in the present study. All cat owners provided consent for the use of data and samples for the present study. The characteristics of the disease were confirmed through comprehensive examination of the apparent clinical symptoms (anorexia, underactivity, vomiting, diarrhea, seizures, tremors, stagger, or others); qualitative PCR-based detection of feline coronavirus in the blood, ascites, or pleural effusions; and laboratory tests [hematocrit (HCT), whole cell count, total protein, aspartate transaminase (AST), alanine transaminase (ALT), total bilirubin (T-bilirubin), albumin-to-globulin ratio (A/G ratio), serum amyloid-A (SAA), and α1-acid glycoprotein (α1AG)]. The age, body temperature, and weight of all cats were recorded at the initial veterinarian’s consultation.

In the collected samples of ascites and/or pleural effusion, the feline coronavirus gene was detected using PCR by Canine Lab Co., Ltd. (Tokyo, Japan), according to the standard technology [20]. The samples were considered positive for feline coronavirus when the target gene was detected in ascites and/or pleural effusion. PCR was performed on blood samples from cats that did not demonstrate ascites or pleural effusion and were candidates for the group of cats without FIP. Cats with samples that tested positive for the viral gene were diagnosed with FIP.

At the time of initial interviews with the cat owners, we measured the parameters of appetite (volume, frequency, and speed of feed intake) and activity (momentum, walking speed, and agility). Then, each parameter was converted to an arbitrary value as an appetite or activity score on a scale of 0–10, and the scores were analyzed statistically to assess the physical conditions at the owner’s home, along with the results of the routine physical examination at the clinic (body temperature, weight, echography, auscultation, and palpation). In addition, SAA levels exceeding the upper limit of the measurable range were treated as 225 μg/mL in the statistical analysis. Furthermore, cats presenting the circulating α1AG levels exceeding the upper limit of the measurable range (736 μg/mL) were considered to be positive [21].

Clinical specimens (EDTA-added whole blood or plasma samples isolated via centrifugation from heparinized whole blood) were collected from all cats at the time of initial medication at our hospital and following the completion of Mutian X treatment (84 days after the first medication). Plasma T-bilirubin levels and A/G ratio were measured using the DRI-CHEM4000V system (FUJIFILM Corporation, Tokyo, Japan), and plasma SAA levels were measured using the DRI-CHEM IMMUNO AU10V system (FUJIFILM Corporation). HCT in the EDTA-treated whole blood samples was measured using MEK-6550 Celltac-α (NIHON KOHDEN Corporation, Tokyo, Japan), and plasma α1AG levels were determined using the latex agglutination method by FUJIFILM VET Systems Co., Ltd. (Tokyo, Japan). PCR of the EDTA-added whole blood, ascites, or pleural effusion samples was performed by Canine Lab Co., Ltd (Tokyo, Japan) [21].

### 2.3. Statistical Analysis

Numerical indicators, including age, appetite score, activity score, body temperature, body weight, HCT, A/G ratio, T-bilirubin, and SAA, were compared between the survivor and non-survivor groups using the Mann–Whitney non-parametric U-test. The presence of diarrhea, vomiting, and FIPV gene expression and significant α1AG levels were classified into categorical data and analyzed using Fisher’s exact tests as 2 × 2 contingency tables with any expected cell value below 5. Additionally, we measured four numerical parameters, including body weight, HCT, A/G ratio, and SAA levels, of the surviving cats treated with Mutian X. Differences in the values before and after drug administration were assessed using Wilcoxon signed-rank test. A *p* value < 0.05 was considered significant. All statistical analyses were performed using StatView 5.0 (SAS Institute, Cary, NC, USA).

## 3. Results

### 3.1. Comparison of Parameters between Cats with and without FIP

The findings in cats with effusive FIP (*n* = 141) or without FIP (*n* = 28) are summarized in Table 1. The age of the cats enrolled in the present study ranged from 1 to 11 years, and approximately 80% of the cats with effusive FIP were below 2 years of age. Effusive FIP was noted in approximately 30% crossbreds and 70% purebreds, including Scottish Fold, Bengal, and Norwegian Forest cats. Among the cats without FIP (*n* = 28), five presented with rhinitis; three with gastroenteritis; two each with pyogenic granuloma, idiopathic uveitis, cerebral infarction, lymphoma, or estrus; one with gastrointestinal eosinophilic sclerosing fibroplasia, hepatic lipidosis, epidermal tumor, idiopathic hyperammonemia, portosystemic shunt, Trichomonas infection, cryptogenic epidermal lymphadenoma, cryptogenic ascites, encephalitis, or lymphadenitis. Of note, cases of encephalitis and lymphadenitis were not considered FIP-associated due to clinical course and resolution of symptoms to conventional therapy.

There were no significant differences in age, body temperature, or weight between cats with and without effusive FIP. However, appetite score (*p* < 0.003), activity score (*p* < 0.002), HCT (*p* < 0.0001), and A/G ratio (*p* < 0.0001) were significantly lower in cats with FIP than in those without it (Table 1). Moreover, the mean T-bilirubin (*p* < 0.002) and SAA (*p* < 0.0001) levels of cats with effusive FIP were significantly higher (over three times) than those of cats without FIP (Table 1).

Consistent with our expectation, the positivity rate of feline coronavirus genes detected using PCR in the blood, ascites, or pleural effusion samples was apparently higher in cats with effusive FIP than those without it. Likewise, the positive rate of α1AG was evidently higher in cats with effusive FIP (125/126, 99%) than in those without it (60%, 15/25); thus, α1AG can indeed serve as a subsidiary indicator for FIP diagnosis, as already proposed previously [1,4]. The prevalence of diarrhea or vomiting was not significantly different between the two groups (Table 2).

### 3.2. Comparison of Parameters between Surviving and Non-Surviving Cats with FIP

In the present study, Mutian X was administered to 141 pet cats with effusive FIP in Japan according to the standard treatment schedule for 84 days. Of these, approximately 82% (116/141) cats survived and their quality of life was substantially improved (Table 3 and Table 4). In majority of the cases, Mutian X was administered orally, and only a few cases received subcutaneous injections. Subcutaneous administration was feasible only during a short period at the early stages of FIP, and it was preferred only in cases where oral administration was difficult due to disease progression; however, treatment could not be continued because of the worsening of symptoms, and almost all such cats died. Therefore, we could not characterize the effects of Mutian X administered via subcutaneous injection due to the small number of cats with worsening symptoms in the present study.

In addition, we explored the differences in age, body temperature, weight, appetite score, activity score, and blood test parameters (HCT, A/G ratio, T-bilirubin, and SAA) at the first examination between the survivor group that received Mutian X treatment for 84 days (*n* = 116) and the non-survivor group that could not complete the therapy (*n* = 25) (Table 3). Ages, body weight, HCT, A/G ratio, and SAA levels were comparable between the two groups. However, body temperature (*p* < 0.0005), appetite score (*p* < 0.0001), and activity score (*p* < 0.0001) prior to the initial drug administration were significantly lower in the non-survivor group than in the survivor group. Interestingly, the T-bilirubin levels were significantly higher in the non-survivor group (approximately three times) than those in the survivor group (*p* < 0.0001) (Table 3).

Furthermore, there were no significant differences in the frequency of diarrhea prior to the initial drug administration between the survivor and non-survivor groups, although vomiting tended to be more frequent in the latter group (*p* < 0.003). Similarly, there were no significant difference in the number of cats with elevated α1AG levels and those positive for feline coronavirus genes in the plasma, ascites, or pleural effusions between the survivor and non-survivor groups (Table 4).

We also noted a few cases of recurrence, presenting with multiple symptoms including appetite loss, activity loss, fever, neurological manifestations, ascites deposition, or pleural effusions, within 4 weeks after the completion of the standard drug course. In such cases, Mutian X was re-administered at an increased dose (200 mg/kg) for 42 days. There were few cases of recurrence (2.1%, 3/141), included in the surviving group as shown in Table 3 and Table 4. The recurrent cases are being followed up, and their outcomes will be analyzed in our future study.

### 3.3. Changes in the Parameters of Surviving Cats with Effusive FIP before and after Mutian X Treatment

Among the surviving cats (*n* = 116), changes in measurable parameters, including body weight, HCT, A/G ratio, and SAA levels, were compared before and after the standard Mutian X therapy (Figure 1). The body weight, HCT, and A/G ratio of the surviving cats were significantly improved after drug treatment compared with their values at the initial examination (*p* < 0.0001). In contrast, SAA levels were drastically decreased after drug treatment (*p* < 0.0001) (Figure 1).

### 3.4. Circulating T-Bilirubin Levels and Survival Rates of Cats with Effusive FIP

The survival rates of cats following the standard Mutian X treatment were calculated based on their circulating T-bilirubin levels. As such, 28 of the 29 cats whose T-bilirubin levels were ≤0.5 mg/dL survived (survival rate, 96.6%). Furthermore, 24 of the 27 cats whose T-bilirubin levels were between >0.5 and ≤1.0 mg/dL (88.9%); 15 of the 20 cats whose T-bilirubin levels were between >1.0 and ≤2.0 mg/dL (75.0%); and 9 of the 18 cats whose T-bilirubin levels were between >2.0 and ≤4.0 mg/dL (50.0%) survived. Meanwhile, only one of the seven cats whose T-bilirubin level was ≥4.0 mg/dL (14.3%) survived.

## 4. Discussion

At present, there is no effective vaccine for FIP, and its mortality rate remains extremely high. Several approaches have been employed to treat cats with FIP, including steroidal and interferon therapies for non-specific immune stimulation with the aim of overcoming the infection [4,5,6,7,8,9]. Some recent in vitro experiments have revealed the potential antiviral therapeutic efficacy of itraconazole, classified as an azole anti-fungal agent, in cats at the early stages non-effusive FIP [10]. Chloroquine—a drug for malaria— inhibited FIPV replication and exhibited anti-inflammatory properties in vitro; however, the in vivo antiviral activity of this drug is inferior to its in vitro activity, leading to toxic effects in the host [11]. Furthermore, at extremely high dosage, cyclosporin, an immunosuppressive drug, inhibited FIPV replication, although this medication produced evident side effects, including gastrointestinal disorders [12,13]. Overall, the currently used drugs do not offer substantial clinical benefits in cats with FIP and warrant further studies for proper control [4,5,6].

The adenosine nucleoside analog GS-441524 has been shown be significantly effective against FIPV, and its therapeutic efficacy against naturally occurring FIP has already been demonstrated [14,16]. According to the package of the commercial drug, MT0901 is the active ingredient of Mutian X. The molecular formula and weight of MT0901 are identical to those of GS-441524, although the chemical structure may slightly differ between the two; therefore, the pharmacological activity of the active ingredient of Mutian X may be similar to that of GS-441524.

Mutian X, a drug product that can be administered orally, has been reported to be markedly effective against spontaneously occurring FIP [22]. Recently, a significant effect of oral administration of Mutian X has been shown on 18 cats with naturally occurring FIP, resulting in recovery with dramatic improvement of clinical and laboratory parameters at the early phase of treatment and no serious adverse effects, and the chemical structure of the active ingredient of Mutian X was identified as that of GS-441524 by means of chemical and structural analysis [23]. This study also showed that the concentration of active ingredient MT0901 in Mutian X tablets can be specified only by the package insert, and additionally mentioned a concern about no approved information on its purity, accuracy, or lot-to-lot variation, because of its position as a veterinary drug unauthorized by a regulatory agency. However, all 18 cats were apparently cured with no critical adverse events detected, supporting that the dose given in the package insert was approximately correct [23]. Thus, the therapeutic efficacy of Mutian X against naturally occurring FIP in 18 pet cats were assessed in this study for the first time. Finding of our current study also confirmed therapeutic effects of Mutian X with better statistical accuracy, on 141 pet cats with FIP under diverse dietary and environmental conditions, consequently revealing statistically significant therapeutic effectiveness for the disease. In the present study, approximately 80% (116/146) of the cats with effusive FIP could be treated effectively and their quality of life could be improved by Mutian X administration (Figure 1, Table 3 and Table 4). Overall, even though the mortality rate of the disease is nearly 100%, majority of the cats with effusive FIP could be treated efficiently in the present study.

The effusive type is the most prevalent form of FIP [1,3]. Thus, we considered it significant to demonstrate the therapeutic efficacy of Mutian X against effusive FIP, accounting for majority of the cases treated at our clinic. Thus, we only included cats with effusive FIP in the present study, because the number of cases of sufficient for statistical analysis (*n* = 141). Nonetheless, the efficacy of Mutian X against non-effusive FIP (including neurological or ocular FIP) has also been demonstrated [22]. Future studies should examine larger cohorts of cats with non-effusive or mixed types, including sufficient number of cases for reliable statistical analysis.

Furthermore, we measured the whole blood cell count, total protein level, and hepatic function parameters (AST and ALT) of all cats enrolled in the study; however, these clinical indicators were excluded from the statistical analyses because no novel significant characteristics could be investigated, and no previous report has described their association with FIP. As such, elevated T-bilirubin levels are not always correlated with higher AST or ALT. Specifically, in cats with FIP, parenchymal liver disease is known to cause hyperbilirubinemia, although increased erythrocyte fragility leading to hemolysis with decreased clearance of hemoglobin-derived products may also be the underlying reason [4,24]. T-bilirubin, A/G ratio, SAA, and α1AG, all of which were assessed in the present study, are correlated with the onset or severity of FIP [3,4,25,26,27,28]. In the present study, almost all cats with effusive FIP showed elevated α1AG levels, exceeding the upper limit of the measurable range of our clinical laboratory system (2200 μg/mL); thus, the precise levels could not be determined. Therefore, we used categorical analysis using Fisher’s exact test to investigate the differences between cats with or without FIP, considering the animals exhibiting α1AG levels exceeding the upper limit of the normal range (736 μg/mL) as positive for the disease [21]. Based on our analyses, multiple pre-dilutions of plasma samples are necessary to accurately quantify α1AG levels because of their drastic increase in cats with effusive FIP.

To assess therapeutic efficacy of Mutian X in cats with effusive FIP and predict the potential treatment outcomes, we examined several clinical parameters of 141 cats at the initial examination and compared the numeral or categorical measures between the surviving cats benefiting from Mutian X treatment and the non-surviving cats refractory to the drug (Table 3). To the best of knowledge, the present study is the first to document elevated circulating T-bilirubin levels in cats with effusive FIP, which possibly correlated with an incurable disease. In addition, based on our statistical analyses, body temperature, appetite score, and activity scores can be used as the possible indicators of survival (Table 3). Previously, in serial blood examinations of cats with effusive FIP, anemia and elevated T-bilirubin were noted in cats from 2 weeks to 0 days before death, and packed cell volume and bilirubin were established as clinical indicators to predict disease staging and survival, providing useful information for the antemortem diagnosis of FIP [29].

Here, we propose a distinct scheme to identify with certainty the potentially rescuable cases of effusive FIP following Mutian X treatment based on the results of laboratory tests at the time of initial examination. In the present study, 96.6% of the cats with effusive FIP whose T-bilirubin levels were 0.5 mg/dL or lower, which is the reference value in healthy cats [21], could be treated effectively. Meanwhile, only 14.3% of the cats with effusive FIP whose T-bilirubin levels were 4.0 mg/dL or higher could be treated effectively; therefore, we must take into account the possibility that Mutian X may not produce a sufficient therapeutic effect in some cases and alternative therapies may be required. Thus, the classification of the disease status of effusive FIP based on the initial T-bilirubin level as the predictor of survival may offer valuable information to the veterinarians and cat owners when making decisions regarding cat treatment. Through veterinarian’s recommendation of Mutian X treatment for their clients, additional prospective benefits of Mutian X therapy in cats with effusive FIP can be expected when the T-bilirubin levels are within the normal range.

In cases from which ascites or pleural effusions could not be collected, we conducted routine qualitative PCR using blood samples, which confirmed that 139 of the 141 (98.6%) cats were FIPV-positive. Based on the results of PCR, the expression of the coronavirus gene was apparently reduced in 116 cats with effusive FIP, which survived following the standard Mutian X treatment; moreover, of these, 112 cats that were positive for FIPV tested negative following the therapy. In addition, body weight, HCT, and A/G ratio were significantly improved following the drug treatment, while the circulating levels of SAA, an acute-phase protein, were drastically reduced, indicating evident improvement in the animals’ quality of life (Figure 1).

Feline coronavirus is a pathogen of the gastrointestinal tract of cats and is typically spread via fecal–oral transmission from the infected animals [1,5]. In a recent study, the minimum and short-term doses of Mutian X ensured viral clearance from the feces of asymptomatic virus-shedding cats, resulting in the establishment of feline corona virus-free cats [30]. A large load of feline coronaviruses can be excreted through the feces of infected cats; on the contrary, in the cats with FIP, the FIPV load excreted through feces is small, suggesting low risk of the horizontal transmission of the pathogen among indoor cats without close contact [31]. In addition, some previous studies have already revealed that the widespread use of anti-viral drugs as preemptive therapy does not contribute to the public benefit, because it certainly increases the risk of multi-drug resistance and epidemic infections [32,33,34]. Given the severity and mortality rates of FIP at onset, the potential epidemics of multi-drug-resistant strains represents a great public threat. Therefore, Mutian X should be administered selectively to animals with FIP for achieving specific therapeutic effects, leading to diverse benefits for the cats and their owners.

A limitation of the present study is that the follow-up period of only 3 months could be set, and it was not possible to investigate the recurrence of FIP thereafter. We usually advise owners to pay close attention to FIP recurrence given the possibility of contact infection from foreign cats, although the possibility of the cats coming into contact with others cannot be excluded. We are currently planning additional follow-up studies of cats surviving for over 3 months following the Mutian X treatment to assess their outcomes and recurrence.

Another limitation is the lack of information about Mutian X components. Since this anti-FIP agent has not been approved by the authorities yet and its official clinical data are lacking, it may be necessary to continue to pay close attention to safety as before. MT0901 is believed to be the main active ingredient, but the efficacy and safety of the other ingredients against FIP have not been elucidated. Any data disclosure from the manufacturer should be required in the future regarding the significance of ingredients other than MT0901. Although Mutian X has been supplied continuously to our hospital by the reliable manufacturer, on the other hand, some inferior products remain to be distributed in the net market. Therefore, cat owners should check the contents of the package insert by themselves, and alternatively, consult with veterinarians having a lot of experience in using genuine Mutian X.

According to a recent report based solely on the results of an Internet-based survey, various GS-441524-like therapies are applied by several owners at homes to effectively abrogate the clinical signs of cats suspected of having FIP, and such uncontrolled treatments appeared to effective against the diverse clinical manifestations of the disease [35]. A significant number of owners also prefer Mutian X drug products; unfortunately, however, almost all of them may receive negligible help from their veterinarians and may use erroneous drug dosage or misinterpret clinical parameters [35]. Thus, it would be ideal for the cat owners to obtain significant help and diagnostic support from veterinarians for ensuring the success of Mutian X therapy. Additionally, the therapeutic effects and outcomes can be predicted accurately by measuring T-bilirubin levels prior to the initiation of Mutian X.

## 5. Conclusions

The present study demonstrated the efficacy of Mutian X in household pet cats with naturally occurring effusive FIP. Almost all cats with effusive FIP (96.6%) whose T-bilirubin levels were below 0.5 mg/dL prior to the treatment could be saved, although the survival rate of cats whose pre-treatment T-bilirubin levels exceeded 4.0 mg/dL remained drastically low (14.3%). Thus, cats with effusive FIP whose circulating T-bilirubin levels are significantly elevated prior to initial drug administration may not satisfactorily respond to or even survive with Mutian X treatment, providing us with reasonable opportunities to undertake specific secondary measures, such as the discontinuation of the drug or the administration of alternative therapies at an earlier stage. Overall, a significant increase in circulating T-bilirubin levels may serve as a prognostic risk factor for severe FIP and may be used a predictor of the clinical benefits of Mutian X therapy.

## Figures and Tables

**Figure 1 vetsci-08-00328-f001:**
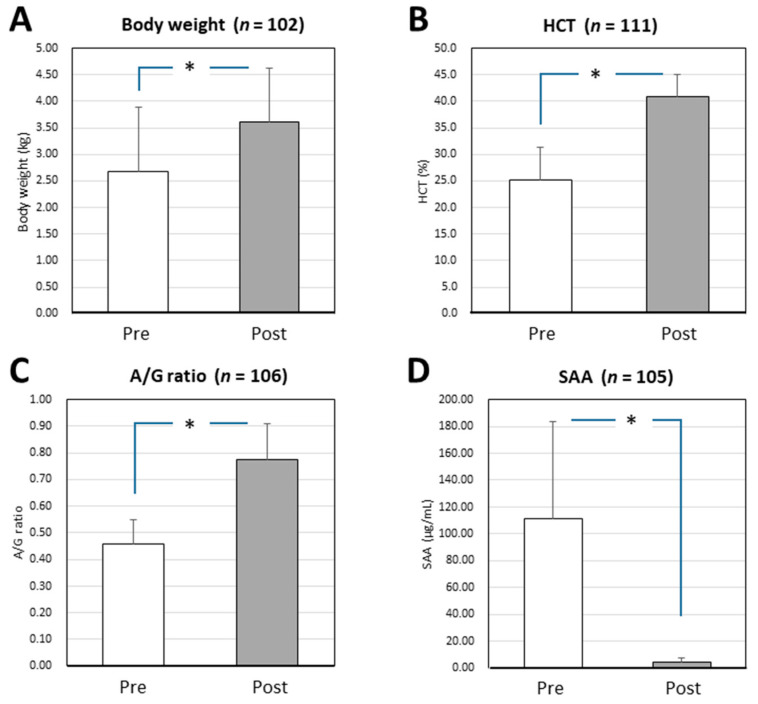
Changes in the parameters of surviving cats with effusive FIP before and after Mutian X treatment. The body weight (**A**), HCT (**B**), and A/G ratio (**C**) of the surviving cats were significantly improved after Mutian X therapy (Post) compared with their values at the initial examination prior to drug therapy (Pre) (* *p* < 0.0001). In contrast, SAA levels (**D**) were drastically decreased after therapy (* *p* < 0.0001). Means and standard deviations are indicated by open/shaded bars and vertical lines, respectively. All statistical analysis were performed using Wilcoxon signed-rank test (non-parametric). HCT, hematocrit; A/G, albumin-to-globulin ratio; SAA, serum amyloid-A.

**Table 1 vetsci-08-00328-t001:** Statistical comparison of signalment and clinical parameters between cats with (*n* = 141) and without (*n* = 28) FIP.

Parameters	Cats with Effusive FIP			Cats without FIP			*p*-Value *
	*n*	Mean	SE	*n*	Mean	SE	
Age (months)	141	15.72	1.92	28	19.36	3.81	NS
Appetite score	141	3.94	0.28	28	6.29	0.69	<0.003
Activity score	141	4.55	0.21	28	6.61	0.68	<0.002
Body temperature (°C)	105	38.80	0.09	23	38.48	0.19	NS
Body weight (kg)	141	2.68	0.10	28	2.68	0.19	NS
HCT (%)	138	25.18	0.54	27	34.45	1.50	<0.0001
A/G ratio	132	0.46	0.01	26	0.70	0.04	<0.0001
T-bilirubin (mg/dL)	101	1.46	0.17	13	0.45	0.16	<0.002
SAA (μg/mL)	130	109.96	6.36	27	33.46	11.44	<0.0001

FIP, feline infectious peritonitis; SE, standard error of the mean; NS, not significant; HCT, hematocrit; A/G ratio, albumin-to-globulin ratio; T-bilirubin, total bilirubin; SAA, serum amyloid-A. * *p* < 0.05, Mann–Whitney non-parametric U-test.

**Table 2 vetsci-08-00328-t002:** Statistical comparison of categorical parameters between cats with and without effusive FIP.

Parameters	Cats with Effusive FIP			Cats without FIP			*p*-Value *
	*n*	Positive	Negative	*n*	Positive	Negative	
Diarrhea	141	22	119	28	7	21	NS
Vomiting	140	10	130	28	5	23	NS
PCR testing (blood)	125	114	11	26	0	26	<0.0001
PCR testing (ascites or pleural) **	141	139	2	2	0	2	<0.0001
α1AG ***	126	125	1	25	15	10	<0.0001

FIP, feline infectious peritonitis; NS, not significant; PCR, polymerase chain reaction; α1AG, α1-acid glycoprotein. * *p* < 0.05, Fisher’s exact test. ** The samples were considered positive for feline coronavirus when the target gene was detected in both or either ascites or pleural effusions. *** The samples were considered positive if the α1AG level exceeded the upper limit of the measurable range (736 μg/mL).

**Table 3 vetsci-08-00328-t003:** Statistical comparison of signalment and clinical parameters between surviving and non-surviving cats with effusive FIP.

Parameters	Survived			Non-Survived			*p*-Value *
	*n*	Mean	SE	*n*	Mean	SE	
Age (months)	116	15.18	2.10	25	17.80	4.76	NS
Appetite score	116	4.42	3.22	25	1.72	2.51	<0.0001
Activity score	116	4.94	2.39	25	2.72	2.48	<0.0001
Body temperature (℃)	86	38.99	0.79	19	37.94	1.21	<0.0005
Body weight (kg)	116	2.70	1.21	25	2.57	1.02	NS
HCT (%)	113	25.11	6.30	25	25.51	6.33	NS
A/G ratio	108	0.46	0.09	24	0.45	0.09	NS
T-bilirubin (mg/dL)	77	0.94	0.95	24	3.11	2.35	<0.0001
SAA (μg/mL)	107	112.19	72.67	23	99.79	72.76	NS

FIP, feline infectious peritonitis; SE, standard error of the mean; NS, not significant; HCT, hematocrit; A/G ratio, albumin-to-globulin ratio; T-bilirubin, total bilirubin; SAA, serum amyloid-A. * *p* < 0.05, Mann–Whitney non-parametric U-test.

**Table 4 vetsci-08-00328-t004:** Statistical comparison of categorical parameters between surviving and non-surviving cats with effusive FIP.

Parameters	Survived			Non-Survived			*p*-Value *
	*n*	Positive	Negative	*n*	Positive	Negative	
Diarrhea	116	17	99	25	5	20	NS
Vomiting	115	4	111	25	6	19	<0.003
PCR testing (blood)	103	92	11	22	22	0	NS
PCR testing (ascites or pleural) **	116	114	2	25	25	0	NS
α1AG ***	104	103	1	22	22	0	NS

FIP, feline infectious peritonitis; NS, not significant; PCR, polymerase chain reaction; α1AG, α1-acid glycoprotein. * *p* < 0.05, Fisher’s exact test. ** The samples were considered positive for feline coronavirus when the target gene was detected in both or either ascites or pleural effusions. *** The samples were considered positive if the α1AG level exceeded the upper limit of the measurable range (736 μg/mL).

## Data Availability

The datasets used and/or analyzed during the current study are available from the corresponding author upon reasonable request.

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
