# Peer review of "Therapeutic Effects of Mutian® Xraphconn on 141 Client-Owned Cats with Feline Infectious Peritonitis Predicted by Total Bilirubin Levels"

_vetsci, 2021, doi:10.3390/vetsci8120328_

Round 1

Reviewer 1 Report

Title – Somewhat long and rambling.  This should be made more cohesive (it’s kind of like 2 different title stuck together).

Significant editing for English language is required – the improper word choice and grammar moderately to sometimes greatly interferes with comprehension.  Re-review will be necessary following English language revisions.  As an example, in line 38, the authors state “…characterized by fibrous thoracic peritonitis…”.  Thoracic peritonitis is, of course, nonsensical.  But I suspect this to be a language barrier issue, rather than a true misunderstanding of the meaning behind this sentence.

Lines 42-46: It is important to mention that diagnosis is large presumptive, based on a conglomeration of clinicopathologic findings, and it is an over-reach to say that “diagnosis…has been preformed comprehensively by analyzing FIP-clinical features” because there really is no specific antemortem test for FIP.  Most of the clinical features are NOT specific to FIP.  Again, this may be a language barrier issue.

Lines 47-55: Authors fail to mention in the introduction any of the previous GS-441524 or GC-376 studies successfully treating FIP in cats (authors include B. Murphy, N. Pedersen, and others).  This is found in the discussion, but should certainly be mentioned in the Introduction, ahead of drugs like interferon and itraconazole.

Lines 62-67: Are the authors implying that the manufacturers of Mutian X do not know the main active ingredient in their product?  Or that it is proprietary?  In the FIP treatment world, Mutian X is known to contain GS-441524.  This should be addressed here and in the discussion (lines 283-288)

Methods – the authors should justify the inclusion of only effusive cases here.

Methods – why not dilute plasma/serum to accurately measure α1AG?

Methods – how were data from relapse cases included (or not) in this study?

Line 149-150: What is meant by simply “neurologic dysfunction” and “inflammatory disease”?  Those are not a diagnosis, and could indeed characterize a cat with FIP.

Results – You should report how many cats received oral vs. SQ treatment, and how this differs among those who survived vs. died.

Discussion – limitations of only 3 month follow-up should be mentioned.

Line 376-387: Authors do not appear to recognize that the reason owners cannot get help from their veterinarians is that the compound found in Mutian X is under patent restrictions in many parts of the world, and is therefore illegal for veterinarians to prescribe or assist with.  They go so far as to recommend that cat owners week help from veterinarians.  But perhaps this is correctly understood by the authors, and simply not communicated appropriately.

In summary, this is a valuable report with some very interesting findings that would be a beneficial contribution to the literature.  However, English language deficiencies result in generally poor readability.  I recommend that the authors seek help from a scientific language editing service.  This is necessary prior to consideration for publication; and further review will be required after editing.

Author Response

October 23, 2021

Dear Academic Editor and Veterinary Sciences editorial team:

Re: Manuscript ID: vetsci-1410022

We thank the Academic Editor and reviewers for their thoughtful criticism and insightful suggestions on the submitted manuscript. Our point-by-point responses to the reviewers’ comments are presented below. For revisions in the article, the corresponding lines and page numbers in the manuscript are indicated. We believe that these amendments have substantially improved our description and hope that the revised manuscript is now suitable for publication in Veterinary Sciences.

We thank you again for your consideration of our work.

Sincerely,

Masato Katayama

Reviewer #1

  1. Title – Somewhat long and rambling. This should be made more cohesive (it’s kind of like 2 different title stuck together).

Response: We have shortened the title without significantly changing its meaning as follows:

Previous title:

“Pharmacological Therapy of 141 Client-Owned Cats with Feline Infectious Peritonitis with Mutian® Xraphconn and Therapeutic Effectiveness Predicted by Circulating Levels for Total Bilirubin”

Shortened title (at the top of page 1 in our revised manuscript):

“Therapeutic Effects of Mutian® Xraphconn on 141 Client-Owned Cats with Feline Infectious Peritonitis Predicted by Total Bilirubin Levels”

  1. Significant editing for English language is required – the improper word choice and grammar moderately to sometimes greatly interferes with comprehension. Re-review will be necessary following English language revisions.  As an example, in line 38, the authors state “…characterized by fibrous thoracic peritonitis…”.  Thoracic peritonitis is, of course, nonsensical.  But I suspect this to be a language barrier issue, rather than a true misunderstanding of the meaning behind this sentence.

Response: The manuscript has now been edited by a professional English editor. We hope that the revised descriptions now meet the required standards of academic English. We have rectified all sentences pointed out by the reviewer (line 38 in the previous manuscript; line 35, page 1, in the revised manuscript).

  1. Lines 42-46: It is important to mention that diagnosis is large presumptive, based on a conglomeration of clinicopathologic findings, and it is an over-reach to say that “diagnosis…has been preformed comprehensively by analyzing FIP-clinical features” because there really is no specific antemortem test for FIP. Most of the clinical features are NOT specific to FIP.  Again, this may be a language barrier issue.

Response: According to the reviewer’s comment, we have corrected the description of diagnosis in the revised manuscript (line 40, page 1 in the revised manuscript). We have replaced the term “diagnosis” with “characteristics” (line 104, page 3 in the revised manuscript).

  1. Lines 47-55: Authors fail to mention in the introduction any of the previous GS-441524 or GC-376 studies successfully treating FIP in cats (authors include B. Murphy, N. Pedersen, and others). This is found in the discussion, but should certainly be mentioned in the Introduction, ahead of drugs like interferon and itraconazole.

Response: We have moved the description of GS-441524 from the Discussion section to the Introduction section (line 54, page 2 in the revised manuscript) and simplified the previous explanation of GS-441524 in the Discussion section (line 279, page 7 in the revised manuscript). Furthermore, we have described the effects of GC-376 on FIP in the same paragraph of the Introduction section (line 61, page 2 in the revised manuscript).

  1. Lines 62-67: Are the authors implying that the manufacturers of Mutian X do not know the main active ingredient in their product? Or that it is proprietary?  In the FIP treatment world, Mutian X is known to contain GS-441524.  This should be addressed here and in the discussion (lines 283-288)

Response: We obtained the package insert of Mutian® Xraphconn (Mutian X) from the commercial product, in which “MT0901” is mentioned as the active ingredient. We confirmed that the molecular formula and weight of MT0901 are identical to those of GS-441524. However, we the chemical structure of MT0901, as shown in the figure in the package insert, is slightly different from that of GS-441524 (CAS No.: 1191237-69-0, reference site indicated in the following section). Unfortunately, we could not obtain any further information on the compound from the manufacturer because of corporate confidentiality. Therefore, apart from the available information, we do not see any other way to explain why the active ingredient of Mutian X is a GS-441524-like drug. We have added the above information to the revised manuscript (line 281, page 7 in the revised manuscript).

GS 441524 | 1191237-69-0 | Biosynth Carbosynth

In addition, a PDF file of the package insert is attached for the reviewer’s reference, and if necessary, it can be submitted as a supplement to our article upon publication (Please refer the attached PDF file entitled as “Package Insert of Mutian Xraphconn”).

  1. Methods – the authors should justify the inclusion of only effusive cases here.

Response: We only enrolled cats with effusive FIP in the present study since a satisfactory number of animals for statistical analysis could be included. Meanwhile, we encountered few cases of non-effusive or mixed FIP at our hospital prior during the study period. We have explained this in the Materials and Methods section (line 97, page 3 in the revised manuscript).

  1. Methods – why not dilute plasma/serum to accurately measure α1AG?

Response: The measurement of α1AG levels was been outsourced to FUJIFILM VET Systems, and pre-dilution of clinical samples could not be performed even as part of optional services. To resolve this issue, we are now planning to collaborate with other laboratories that offer optional pre-dilution services for our further analyses. In the present study, we used categorical analysis to reveal the significant elevation of α1AG levels in cats with effusive FIP.

  1. Methods – how were data from relapse cases included (or not) in this study?

Response: We have already included a few cases of recurrence (relapse), which we have already described in the submitted manuscript (line 229, page 6 in the previous manuscript). There were only three recurrent cases (3/141). We hope to follow-up these cases further and investigate their outcomes in a future study. These points were presented in the revised manuscript (line 234, page 6 in the revised manuscript)

  1. Line 149-150: What is meant by simply “neurologic dysfunction” and “inflammatory disease”? Those are not a diagnosis, and could indeed characterize a cat with FIP.

Response: In response to the reviewer’s indication regarding the designation of cat disease, we have reviewed our medical records of each case. Accordingly, we have amended “neurological symptom,” “neurological dysfunction,” and “inflammatory disease” in the previous manuscript as “cerebral infarction,” “encephalitis,” and “lymphadenitis” in the revised manuscript, respectively (line 159 and 162, page 4 in the revised manuscript).

  1. Results – You should report how many cats received oral vs. SQ treatment, and how this differs among those who survived vs. died.

Response: In the present study, Mutian X was administered orally in most cases, and its subcutaneous injection was administered in only a few cases (n = <10). Subcutaneous administration was feasibly only during a short period at the early stages of FIP onset and only in cases where oral administration was difficult due to disease progression; however, the treatment cannot be continued thereafter due to the worsening of symptoms, and almost all these cats died.

To accurately compare the therapeutic effects of oral and subcutaneous administration, it is necessary to randomly divide a relatively large number of cats with similar physical conditions into two groups. Under the conditions of the present study, we judged it impossible to characterize the effects of subcutaneous administration in such a small number of cases with worsening symptoms. This point was already partially described in the Materials and Methods section of our previous manuscript, and some further explanation has been added to the revised version (line 195, page 5 in the revised manuscript).

  1. Discussion – limitations of only 3 month follow-up should be mentioned.

Response: We have described the limitations of the study in the revised version (line 369, page 9 in the revised manuscript). As such, a follow-up period of only 3 months could be set, and the recurrence of FIP could not be evaluated thereafter. We plan to conduct follow-up studies on the surviving cats and describe their outcomes in our next report.

  1. Line 376-387: Authors do not appear to recognize that the reason owners cannot get help from their veterinarians is that the compound found in Mutian X is under patent restrictions in many parts of the world, and is therefore illegal for veterinarians to prescribe or assist with. They go so far as to recommend that cat owners week help from veterinarians.  But perhaps this is correctly understood by the authors, and simply not communicated appropriately.

Response: We thank the reviewer for their useful advice. We agree with the reviewer’s point that drug patents may limit veterinarian activities in some cases. However, we have avoided this point, as this represents a subsidiary issue in the context of the present study. Instead, we have revised the description (It would be highly recommended that the cat owners should seek …) in the previous version as “ Thus, it would be ideal for the cat owners to obtain…” in the revised manuscript (line 383, page 9 in the revised manuscript).

  1. In summary, this is a valuable report with some very interesting findings that would be a beneficial contribution to the literature. However, English language deficiencies result in generally poor readability.  I recommend that the authors seek help from a scientific language editing service.  This is necessary prior to consideration for publication; and further review will be required after editing.

Response: We deeply appreciate your understanding of the importance of our study. As described in the previous section, we have used a professional English editing service by a native editor for language correction.

Reviewer 2 Report

See attachment

Author Response

October 23, 2021

Dear Academic Editor and Veterinary Sciences editorial team:

Re: Manuscript ID: vetsci-1410022

We thank the Academic Editor and reviewers for their thoughtful criticism and insightful suggestions on the submitted manuscript. Our point-by-point responses to the reviewers’ comments are presented below. For revisions in the article, the corresponding lines and page numbers in the manuscript are indicated. We believe that these amendments have substantially improved our description and hope that the revised manuscript is now suitable for publication in Veterinary Sciences.

We thank you again for your consideration of our work.

Sincerely,

Masato Katayama

Reviewer #2

This reviewer’s comments were added as long sentences without items; here, we have divided them in a numerical list and responded to each point.

  1. This paper has useful data that deserves to be published. The authors have a large cohort of cats with effusive FUIP from Japan that were treated with Mutian X, with an 80% success rate. There is some discussion of the value of bilirubin concentrations as a prognostic indicator. There are some minor quibbles. The authors first language is Japanese, and the English expression is clunky. The use of a professional proof-reading service would improve this aspect.

Response: We thank the reviewer for understanding the significance of our study, which provides useful data, possibly to be published. As described in our responses to reviewer #1, we have used a professional English editing service by a native editor for language correction. We hope the descriptions in the revised version now meet the standards of academic English.

  1. WE need better details of the PCR used for diagnosis.

Response: PCR analysis was outsourced to Canine Lab, a CRO in Japan, and the technology and detailed procedure cannot be disclosed due to corporate confidentiality. In response to the reviewer’s suggestion, we requested the organization to disclose technical details, but we obtained no information other than the published report recommended by the company (Reference No.20: Tanaka. Y, et al. BMC Vet. Res. 2015, 11, 57, as cited the revised manuscript, at line 463 of page 11). We understand that the results of PCR analysis are subsidiary data and have very little involvement in the main issue of the present study. We hope that the current information is sufficient to communicate the main purpose of the research.

  1. The introduction is not good and needs material to be moved from the Discussion to explain how treatments for FIP have developed.

Response: Reviewer #1 gave us the same suggestion. Accordingly, we have moved the description regarding GS-441524 from the Discussion section to the Introduction section (line 54, page 2 in the revised manuscript) and simplified the previous explanation of GS-441524 in the Discussion section of the revised manuscript (line 279, page 7 in our revised version). We have also described the effects of GC-376 on FIP in the same paragraph of Introduction (line 61, page 2 in the revised manuscript).

  1. But there is a central problem – and it’s incredibly important. We don’t know anything about Mutian X. And this is a problem with the small number of papers concerning this drug published by Diane Addie and others. WE have no idea what is inside! Most people believe Mutian contains the adenosine nucleoside analogue GS-441524. Some people say it contains a drug LIKE GS-441524.

Response: Reviewer #1 gave us similar comments.

Accordingly, we obtained the package insert of Mutian® Xraphconn (Mutian X) from the commercial product, in which “MT0901” is mentioned as the active ingredient. We confirmed that the molecular formula and weight of MT0901 are identical to those of GS-441524. However, we the chemical structure of MT0901, as shown in the figure in the package insert, is slightly different from that of GS-441524 (CAS No.: 1191237-69-0, reference site indicated in the following section). Unfortunately, we could not obtain any further information on the compound from the manufacturer because of corporate confidentiality. Therefore, apart from the available information, we do not see any other way to explain why the active ingredient of Mutian X is a GS-441524-like drug. We have added the above information to the revised manuscript (line 281, page 7 in the revised manuscript).

GS 441524 | 1191237-69-0 | Biosynth Carbosynth

In addition, a PDF file of the package insert is attached for the reviewer’s reference, and if necessary, it can be submitted as a supplement to our article upon publication (Please refer the attached PDF file entitled as “Package Insert of Mutian Xraphconn”).

  1. The company website https://www.mutianstore.com/ says “Mutian is a dietary supplement exclusively designed for cats with Feline Infectious Peritonitis (FIP), by boosting their immune system and overall well-being.” In order for this work to be published, the authors need to get a vial of Mutian X and a capsule and have them analysed at an analytic laboratory. We need a validated list of ingredients. Does it contain GS-441524? What is the analytic purity? What other things have been added.

Response: We reasonably agree that each of the issues pointed out by reviewer #2 (analysis of the active and inactive ingredients of Mutian X and validation of their chemical composition, purity, and any other additives) is very important from the perspective of many pharmacological researchers, and we would definitely like to address them when technical and financial resources are available.

As clinical veterinarians encountering cats that die from FIP every day, the authors are devoted to exploring the routine medical practices that can be performed to rescue them, rather than the said pharmaceutical analyses.

According to the descriptions in the package insert of the product currently available on the market, the active ingredient shows a GS-441524-like structure, as explained in the previous section. This package insert is attached for the reviewer’s reference, and it can be submitted as a supplement to the article if considered necessary by the editor or reviewers.

  1. The dosage used in this paper – 100 mg/kg – 100 mg of WHAT per kg? I am confident the drug works. I have no idea if there is batch to batch variation in terms of the active ingredient.

Response: Unfortunately, the above-mentioned package insert contains information on the chemical structure of the active ingredient (MT0901) but no description of its contents. We also recognize that the content of this active ingredient is indispensable for comparison with other reported data; thus, we speculate from the information provided by the manufacturer that 100 mg of the formulation contains approximately 5 mg of the active ingredient. We have included the estimated active ingredient content in the revised manuscript (line 85, page 2 in the revised manuscript).

  1. But I am 100% sure that the manufacturers are not providing sufficient information for us to make any comparison with other formulations of GS-441524 which are available on internet. Mutian X is far more expensive than the other products used to treat FIP, such as Capella, Spark etc. Samantha Evans paper and the amazing resources on Neil’s Pedersen’s website https://sockfip.org/2021-treatment-with-oral-formulations-of-gs-441524/ provide detailed information on how to use GS and the different formulations available, and the benefits of injections over oral formulations in terms of bioavailability. But the knowledge gap is – what is really inside? What concertation is present? Are other active drugs included. This paper will only be publishable if the authors can determine this and provide a validated analysis of the product, they are giving the cats.

Response: As both reviewers have pointed out, we agree that information on Mutian X provided by the manufacturer is insufficient; in response to the Japanese veterinarians’ requests, the manufacturer has made considerable efforts to provide certain information necessary for routine FIP treatment. Accordingly, we have included the chemical structure, molecular formula, and molecular weight of the active ingredient of Mutian X (MT0901) and its estimated content based on the information obtained from the manufacturer’s Facebook page. Regarding its further chemical or pharmaceutical properties, we are waiting for academic reports from scientific researchers in the developing company (Mutian Life Sciences Ltd.), because the rights to publish the information contained in the package insert are reserved to them.

As described on the Facebook page, Mutian X contains 5 mg·of active ingredient in 100 mg of pharmaceutical ingredient; we have added these details to the text. Since Mutian X is an unapproved drug, information on prescriptions other than this (components other than the active ingredient and their contents) are to be released as the final application information after approval by authorities as a veterinary medicine.

In Japan, over 10 veterinary hospitals, including our clinic, are officially recognized by the manufacturer as “MUTIAN Cooperating Animal Hospitals,” and we would like to participate in the effort toward the approval of Mutian X as a veterinary drug.

  1. It is also impossible ethically to use a product that has not been studied in terms of analytic purity and no ethics committee would give permission to conduct this research with a product of unlock purity.

Response: We have already mentioned in the previous manuscript that ethical review and approval were waived for the present study because all data were obtained within the scope of usual veterinary care and properly anonymized (line 404, page 10 in our revised manuscript). We also obtained consent from each owner regarding the acquisition and use of medical data (line 103, page 3 in the revised manuscript) and confirmed that this practice complies with the Declaration of Informed Consent of the Japan Veterinary Medical Association (line 406, page 10 in the revised manuscript).

  1. So, I would look very favourably on the data presented here, but the authors need to do some extra work and work out.

Response: We thank the reviewer for their appreciation of the presented work.

  • is GS-441524 present in Mutian X

Response: As already described earlier, there is no other way to explain why the active ingredient of Mutian X is a GS-441524-like drug.

  • At what concentration?

Response: As already described earlier, based on information from the manufacturer, we speculate that 100 mg of formulation contains 5 mg of the active ingredient.

  • How much variation in the concentration is there from batch to batch?

Response: Unfortunately, there is no information regarding this pharmaceutical aspect.

  • Are other drugs such as liver tonics or preservatives present in the product

Response: Unfortunately, there is no information on the components of the product.

  • What is the excipient used.

Response: According to information in the package insert, the inactive ingredient include microcrystalline cellulose, polyvinylpolypyrrolidone, and magnesium stearate.

  • How much GS-441524 is present in the tablets

Response: As already described earlier, the estimated content of the active ingredient MT0901 and its GS-441524-like structure have been described in the revised manuscript (line 85, page 2 and line 281, page 7, respectively in the revised manuscript).

Round 2

Reviewer 1 Report

English language and readability are improved.

I thank the authors for their comments to my suggestions.

Additional specific points:

  • Line 12 – “the predictive identification of rescuable FIP” is confusing wording. Maybe “predictors of therapeutic success remain unknown” instead?
  • Line 14 – the term “rescued” is still somewhat vague. I would replace throughout with terms like “survived” or “lived” or “treated effectively”
  • Line 15 – what is “owner inquiry-based signalment”? If you mean to imply that owners were asked about age and sex/neuter status, you can just say ‘signalment’ b/c it is very commonplace to get this information from owners.
  • Line 38 – should be “OR ocular manifestations” not “and”
  • Line 39-40 – no non-invasive confirmatory test is available for any type of FIP, effusive or not (although characteristic effusion does make the diagnosis much more likely)
  • Line 45 – it is incorrect to say that there is currently no effective therapy, as you know, many people are doing this at home (see Jones et al 2021 and others). Thus, this should be reworded as “no widely available and legal effective therapy” or similar.
  • Line 56 – should be “the phosphoramidate prodrug OF GS-44154”(b/c GS-5734 is the prodrug, not GS-441523)
  • Line 110 – please include in parentheses the alternate names that will be better recognized in North America (AST and ALT)
  • Line 117-119 – why were cats without ascites or pleural effusion mentioned here? I thought non-effusive cases were not included in the study?
  • Line 127-129 – are you implying that you considered alpha-1AG levels to be diagnostic for FIP…?
  • Line 163 – I still do not understand why cases of encephalitis or lymphadenitis were determined to be cases without FIP. Those two entities can indeed be a part of FIP presentation.  So how was it determined that they did NOT have FIP?
  • Line 222- it looks like vomiting was more frequent in the non-survivor group. Would this be the “later” group (not the “former” group)?
  • Line 236-241 – thank you for the additions here. But please make it clear, one way or the other, whether data from these cats are included in the tables/analysis here
  • Line 301 – should say “including neuro or ocular FIP” b/c there are cases of non-effusive FIP that have neither neuro nor ocular signs (ex: granulomas in kidney and liver)
  • Line 327 – I do not think the elevated Tbili is what caused therapy not to work; rather it is probably an effect of more severe disease. Thus I would change this to “which possibly correlated with incurable disease”.  This is probably a correlation, rather than causation.

Reviewer 2 Report

The authors have done a lot of work. Their reply is polite and helpful, but not helpful enough.

You cannot publish a paper about an unlincensed product purchased on the black market, unless you do a great deal of research about what is in theproduct and how reliable is the formulation.

I have  for my own research  had a tablet of Mutian X assayed - and a tabket that is suppose to contain 10 mg of GS-441524 like material actually has 18 mg of GS-441524. I don’t know if this is huge batch to batch variation, or they just put in more active drug than what is on the label.

But I have no confidence this work could be reproduced by another investsigator - because i don’t have any faith in the product.

So, I have to reject the paper.

There are analytic labs who can assay what substances are in Mutian using mass spectroscopy and GS-HPLC. We need this data, not just for one tablet, but for many tablets from different batches. And the same fr the liquid used for injections.

And then we can compare Mutian to Capella and Spark and other formulations containing GS-441514.

It would not be scientifically valid to publish the manuscript without such data.

Author Response

Thank you for giving us valuable comments. We hope you to confirm the Academic Editor’s opinion about our current report. We also revised our previous manuscript according to the Academic Editor's and another Reviewer's comments. Thanks again for reviewing our article.